# Structural features of Dnase1L3 responsible for serum antigen clearance

Jon J. McCord [1], Minal Engavale[2], Elahe Masoumzadeh[3], Johanna Villarreal[1], Britney Mapp[2], Michael P. Latham [3], Peter A. Keyel [2] & R. Bryan Sutton [1]✉

Autoimmunity develops when extracellular DNA released from dying cells is not cleared from serum. While serum DNA is primarily digested by Dnase1 and Dnase1L3, Dnase1 cannot rescue autoimmunity arising from Dnase1L3 deficiencies. Dnase1L3 uniquely degrades antigenic forms of cell-free DNA, including DNA complexed with lipids and proteins. The distinct activity of Dnase1L3 relies on its unique C-terminal Domain (CTD), but the mechanism is unknown. We used multiple biophysical techniques and functional assays to study the interplay between the core catalytic domain and the CTD. While the core domain resembles Dnase1, there are key structural differences between the two enzymes. First, Dnase1L3 is not inhibited by actin due to multiple differences in the actin recognition site. Second, the CTD augments the ability of the core to bind DNA, thereby facilitating the degradation of complexed DNA. Together, these structural insights will inform the development of Dnase1L3-based therapies for autoimmunity.

[1] Texas Tech University Health Sciences Center, Dept of Cell Physiology and Molecular Biophysics, Lubbock, TX, USA. [2] Texas Tech University, Dept. of Biological Sciences, Lubbock, TX, USA. [3] Texas Tech University, Dept. of Chemistry & Biochemistry, Lubbock, TX, USA. ✉email: roger.b.sutton@ttuhsc.edu

Production of antibodies targeting cell-free nucleic acids can lead to the development of severe autoimmune disorders, including systemic lupus erythematosus (SLE)[1], hypocomplementemia urticarial vasculitis syndrome (HUVS)[2], and rheumatoid arthritis (RA)[3]. The enzymes responsible for clearing antigenic DNA from serum have been identified as Dnase1 and Dnase1L3 (deoxyribonuclease 1 like 3 or Dnaseγ)[4,5]; yet, the activities assigned to these two enzymes are not synonymous. For example, mutations in Dnase1L3 have been linked to multiple autoimmune diseases[2,6–8], while Dnase1 is unable to rescue these autoimmune phenotypes[6,7]. Therefore, Dnase1L3 accounts for a broader range of nucleic acid degradative activities.

Based on the overall sequence conservation between the two enzymes, the catalytic activities of Dnase1 and Dnase1L3 should be similar. Dnase1 and Dnase1L3 are both $Ca^{2+}/Mg^{2+}$-dependent endonucleases with a high degree of identity in the active site; however, the catalytic mechanism of the Dnase1-family remains contested[9]. Currently, there are two leading models for Dnase1 family catalysis, the single cation carboxylic acid model[10] and double divalent cation model[11]. The principal differences between the proposed mechanisms are the number of divalent cations involved in catalysis, and the residues that directly participate in acid/base catalysis. The more recent single cation carboxylic acid model posits a single divalent cation and Asp-189 as the catalytic base[10,12]. In contrast, the more established double divalent cation model[13] predicts two cations and implicates His-155 as the catalytic acid and His-274 as the catalytic base[11]; however, neither model is completely supported by experimental data.

While the catalytic mechanism may be similar, there are multiple functional differences between Dnase1 and Dnase1L3. Dnase1 is inhibited by actin, while Dnase1L3 is not[14]. Furthermore, Dnase1L3 degrades DNA complexed with proteins and lipids, including immune complexes, more efficiently than Dnase1[6,8,15,16]. This unique Dnase1L3 activity is hypothesized to originate from its positively charged C-terminal domain (CTD) that extends from Ser-283 to Ser-305[6]. Previous homology modeling, including models generated by AlphaFold[17], predicts that the CTD forms a rigid α-helix, albeit with low confidence, which projects out from the core enzyme. It has been suggested that the C-terminal α-helix facilitates membrane binding, penetration or could potentially displace bound histones that occlude substrate DNA[6].

Here we investigated the structure of the Dnase1L3 core domain using X-ray crystallography and the isolated CTD using Nuclear Magnetic Resonance (NMR) spectroscopy. The two domains were combined using Small-Angle X-ray Scattering (SAXS). Our high-resolution crystal structure of Dnase1L3 provides the first experimental support for the two-cation catalytic mechanism, wherein His-155 and His-274 act as the catalytic acid and base residues, respectively. The functional effects of the CTD were studied using enzymatic assays, co-sedimentation, and fluorescence polarization. We tie these functional assays to structural features of the CTD with circular dichroism and molecular dynamics to present a model of Dnase1L3's diverse DNA degradation capacity and the function of the intrinsically disordered CTD. Our results support a DNA binding role for the CTD that confers increased avidity to heterogeneous DNA.

## Results

**MBP fusion enables prokaryotic expression of active Dnase1L3.** Dnase1 and Dnase1L3 are nonspecific endonucleases that induce chromosomal DNA laddering. The DNA digesting activity of Dnases complicated previous attempts to overproduce the enzyme using prokaryotic expression systems[18]; however, the expression of inactive Dnase1 mutants has been successful[19]. To produce active Dnase1L3 in bacteria, we expressed the enzyme as a His$_6$-Maltose Binding Protein (MBP)-Dnase1L3 fusion protein. The fusion protein approach facilitates the purification of active enzyme as the Dnase1L3 fusion is inactive prior to tobacco etch virus (TEV) protease cleavage of MBP from Dnase1L3 (Fig. S1a). Prokaryotic expression offers a higher yield and a more straightforward workflow compared to the current eukaryotic expression protocols[20].

**Structure of the core enzyme of Dnase1L3.** Full-length Dnase1L3 was refractory to crystallization, likely because of the flexible C-terminus; therefore, we expressed and purified a form of Dnase1L3 missing the last 23 residues (ΔCTD) for the structural analysis of the core domain. Dnase1L3 ΔCTD was crystallized at pH 8.5 in 250 mM $MgCl_2$ in the P1 triclinic space group with four Dnase1L3 ΔCTD molecules in the asymmetric unit. The core Dnase1L3 enzyme was refined to a final resolution of 2.2 Å with an R-free of 23% (Table 1). The crystal structure of Dnase1L3 exhibits the classic 4-layer α-β sandwich architecture similar to other members of the exo/endo phosphatase (EEP) family of enzymes. The Dnase1L3 core enzyme structure superimposes with human Dnase1[10] with an RMSD of 1.8 Å across all C-α atoms (Fig. S2a). Post-translationally Dnase1 possesses two N-linked glycosylation modifications that are implicated in Dnase1 stabilization and wild-type nuclease activity[21]; however, the N-linked glycosylation sites at Asn-18 and Asn-106 on Dnase1 correspond to Asp residues (Asp-38 and Asp-126) in human Dnase1L3 (Fig. S2b, c).

**Table 1 Crystallographic data and refinement summary.**

|  | Dnase1L3 ΔCTD (S283X) |
|---|---|
| Wavelength (Å) | 1.127 |
| Resolution Range (Å) | 32.06–2.22 (2.30–2.22) |
| Space Group | P1 |
| Unit Cell (Å) |  |
| a, b, c (Å) | 52.31, 72.67, 73.170 |
| α, β, γ (°) | 85.39, 80.56, 88.39 |
| Total reflections | 454844 (18459) |
| Unique reflections | 49718 (4702) |
| Multiplicity | 9.1 (3.9) |
| Completeness (%) | 93.95 (79.91) |
| Mean I/σ(I) | 8.59 (1.01) |
| Wilson B-factor | 51.22 |
| $R_{merge}$ (%) | 14.51 (128) |
| $CC_{1/2}$ | 0.998 (0.526) |
| Reflections used in refinement | 49177 (4189) |
| Reflections used for R-free | 2218 (189) |
| $R_{work}$ (%) | 19.06 (50.29) |
| $R_{free}$ (%) | 23.04 (52.09) |
| Number of non-hydrogen atoms | 8915 |
| macromolecules | 8729 |
| ligands | 12 |
| Solvent | 174 |
| Protein residues | 1052 |
| RMS (bonds, Å) | 0.015 |
| RMS (angles, °) | 1.74 |
| Ramachandran favored (%) | 95.40 |
| Ramachandran allowed (%) | 4.60 |
| Ramachandran outliers (%) | 0.00 |
| Rotamer outliers (%) | 4.27 |
| Clashscore | 4.08 |
| Average B-factor (Å²) | 64.95 |
| macromolecules (Å²) | 65.19 |
| ligands (Å²) | 55.39 |
| solvent (Å²) | 53.89 |

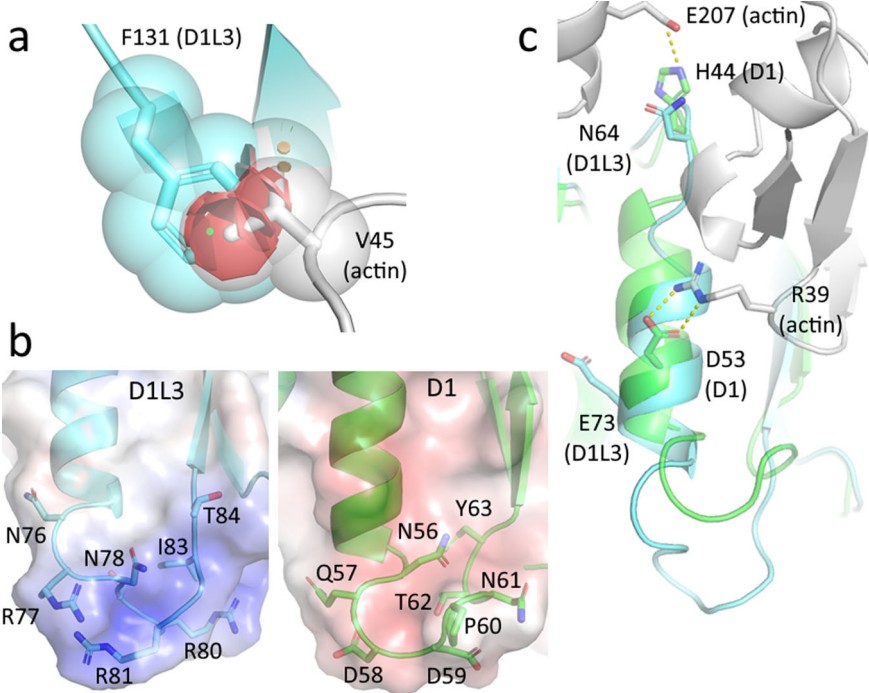

**Fig. 1 Dnase1L3 inhibits actin binding. a** Phe-131 of Dnase1L3 (blue) clashes (in red) with Val-45 of actin (grey) when actin is docked to Dnase1L3[24].
**b** Dnase1 (D1, green) is acidic (pI = 3.45) at the actin interface, whereas Dnase1L3 (D1L3, blue) is basic (pI = 11.15) at the homologous site. **c** The salt bridge interactions from Dnase1 (green) to actin (grey) are disrupted in Dnase1L3 due to the homologous α-helix (residues 67–75) from Dnase1L3 (blue) being out of register relative to the Dnase1 actin-binding α-helix.

**Multifactorial actin resistance of Dnase1L3.** Dnase1 activity is inhibited by G-actin, DNA-actin complexes, and DNA in aggregates with other poly-cations[22]. Consequently, native Dnase1 as a therapy for cystic fibrosis (CF) pulmonary sputum fluidization is limited. Dnase1 was reengineered as Pulmozyme® to minimize actin inhibition, but there remains a weak inhibition of Dnase1 by large DNA/actin complexes in CF patient sputum[23]. Dnase1L3 is natively resistant to actin inhibition[14], making this enzyme a valuable potential therapeutic. We compared Dnase1L3 with the Dnase1-actin crystal structure[24] to elucidate the differences in the actin-binding interface. There are four notable differences between the two enzyme interfaces that directly impact actin binding. First, the 'central hydrophobic' region that Dnase1 utilizes for actin binding is sterically disrupted in Dnase1L3[25]. Dnase1 actin resistance was experimentally conferred by mutating Ala-114 within the 'central hydrophobic region' to a bulky residue such as Arg, Glu, Met, Tyr, or Phe[22]. In Dnase1L3, the homologous residue to Ala-114 is Phe-131, which contributes to its native actin resistance (Fig. 1a). Second, a two amino acid insertion in the loop (residues 75-85 in Dnase1L3) following the actin-binding α-helix creates a more flexible loop in Dnase1L3, with an inverted net charge, relative to Dnase1 (residues 55–63), from a pI of 3.45 in the actin-inhibited Dnase1 structures[24] to pI of 11.15 in the actin-resistant Dnase1L3 (Fig. 1b). The charge inversion in the homologous actin-associated loop likely repels the similarly charged actin from productively binding to Dnase1L3. Third, an important salt bridge forming residue for Dnase1-actin binding is Glu-69 in Dnase1, which associates with Lys-61 from actin[24]. In Dnase1L3, Glu-69 is analogous to Ser-91, which has less potential for a salt-bridge to actin (Fig. S3). Finally, the actin-binding α-helix (residues 67–75) is 'out-of-register' in our crystal structure of Dnase1L3 relative to the homologous α-helix in Dnase1 (Fig. 1c). It is possible that this 'out-of-register' shift disrupts a critical actin binding salt bridge at Glu-73.

**The patient mutation R206C disrupts a critical salt-bridge network.** The best characterized pathogenic mutation in Dnase1L3 is the hypofunctional R206C that reduces enzyme activity and protein secretion[26,27]. Using our crystal structure as a starting point, we performed 500 ns molecular dynamics simulations in explicit solvent equilibrated in the presence of 150 mM $MgCl_2$ on Chain C of Dnase1L3 ΔCTD computationally mutated to R206C. In addition to the previously predicted Glu-170 to Arg-206 salt bridge[28], our structural analysis shows that Arg-206 forms an extensive salt-bridged network with Asp-166 and Glu-167. The crystal structure shows well-ordered electron density for the Arg-206 residue, as well as the salt-bridged Asp-166 residue (Fig. 2a). Glu-167 and Glu-170 are likely transiently salt-bridged with Arg-206 because their electron density is not well-defined. The R206C mutation eliminates the salt-bridge network (Fig. S4) (Supplementary Data 1 and 2). We attempted to express and purify the R206C mutant, but the mutant proved unstable when cleaved from MBP using TEV protease. The fusion protein, Dnase1L3-MBP, was used instead to quantitate the protein misfolding via ANS (8-anilino-1-naphthalenesulfonic acid) fluorescence. The fluorescence of ANS varies as a function of its dielectric environment. The dye demonstrates greater fluorescence in hydrophobic environments, and lesser florescence in aqueous environments; therefore, it has been used to assess changes in protein folding stability[29]. In the MBP-Dnase1L3 constructs, the ANS fluorescence is more intense in the presence of the R206C fusion protein compared to the WT fusion protein or MBP alone. This result indicates that the Dnase1L3 R206C hydrophobic core is more accessible to ANS relative to the WT protein (Fig. 2b) (Supplementary Data 3). Based on molecular dynamics simulation and Rosetta energy scoring[30], the loss of the salt bridge network increases ΔΔG of folding by 14 kcal/mol (Fig. 2c) (Supplementary Data 4). The less favorable free energy of folding of R206C corresponds to the disruption of three salt bridges

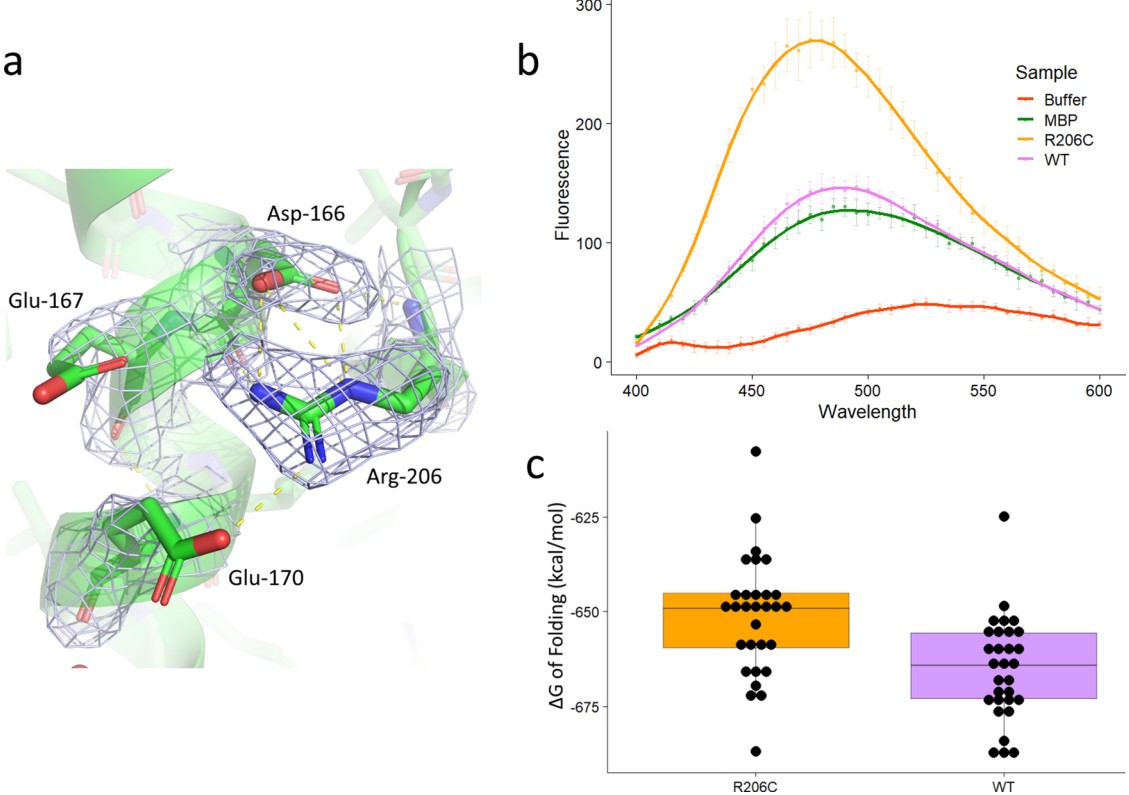

**Fig. 2 The R206C mutation disrupts protein folding. a** Electron density of Dnase1L3 ΔCTD chain C showing the salt-bridged network of Arg-206 joining two helical elements together. The distances indicated are 2.9 Å from Asp-166 to Arg-206, and 3.5 Å and 3.1 Å from Glu-167 and Glu-170, respectively. **b** Mean ANS (8-anilino-1-naphthalenesulfonic acid) fluorescence for buffer, MBP (maltose-binding protein) alone, the Dnase1L3 WT-MBP fusion protein, and the Dnase1L3 R206C-MBP fusion protein. The error bars represent the mean from $n = 3$ independent measurements ± the standard deviation for each data point (**c**). Estimated ΔG of folding in kcal/mol of the WT enzyme and R206C, from 10 snapshots (10 ns apart) of the last 100 ns of each of the six simulations (three for WT and three for R206C, $n = 30$ independent calculations of free energy of folding). The free energy of folding was estimated using Rosetta. The difference between WT and the R206C mutation is 14.03 kcal/mol ± 3.79 (standard deviation), $p = 0.000476$, $n = 30$ independent calculations of free energy for WT and R206, each. The whiskers of the boxplot extend to the farthest data point that is no further than 1.5 times the interquartile range of the data.

(3–5 kcal/mol each)[31], in accordance with our crystal structure. The triple salt-bridged Arg-206 residue is a key stabilizing residue in the Dnase1L3 molecule. Taken together, the reduced structural stability of R206C results in enzyme misfolding, and therefore reduces catalytic activity and impairs secretion to account for the net hypofunctionality of the R206C mutation.

**A single structural divalent cation and a unique disulfide in the core enzyme.** All members of the Dnase enzyme family require both structural and catalytic divalent cations. Dnase1 crystal structures[10,24] show evidence for four $Ca^{2+}$ or $Mg^{2+}$ binding sites, but a total of five cations have been postulated[32]. In Dnase1L3, we observed three divalent cations, with two $Mg^{2+}$ present within the catalytic site (Fig. 3a). Using the nomenclature from Guéroult[32], the 'II' and 'III' structural cations are absent in Dnase1L3. In Dnase1, the 'II' divalent cation binding site is stabilized by a nearby disulfide bridge (Cys-101 to Cys-104). The divalent cation coordinating residues in the 'II' binding site are conserved between Dnase1 and Dnase1L3, but the divalent cation binding site in Dnase1L3 is missing both disulfide cysteines and is one residue shorter relative to the homologous Dnase1 site (Fig. 3b). The missing disulfide bridge in Dnase1L3 prevents the coordinating residues from folding into the consensus divalent cation binding site. The lack of the 'II' cation binding site in Dnase1L3 increases the flexibility of the corresponding loop. Indeed, the B-factors are elevated between the site 'II' loops across

each chain in the Dnase1L3 asymmetric unit (Fig. S5). Importantly, the absence of site 'II' does not disrupt the position of the key DNA binding residue Arg-132 (Arg-111 in Dnase1)[19].

Cation coordination of the 'III' site is disrupted in Dnase1L3 because it lacks a critical $Ca^{2+}$ coordinating residue. The primary $Ca^{2+}$ coordinating residue in Dnase1, Asp-172, corresponds to Gly-193 in Dnase1L3 (Fig. 3c). The conserved conformation of the catalytic residue Asn-191 (Asn-170 in Dnase1) demonstrates that the Dnase1L3 Cys-194 to Cys-231 disulfide bridge may be sufficient for active site stability, even without $Ca^{2+}$ coordination at site 'III'. Additionally, Dnase1L3 includes an extra Cys-24 to Cys-52 disulfide bond, corresponding to Ala-4 and Tyr-32 in Dnase1, respectively. The additional disulfide bridge in Dnase1L3 may partially compensate for the lack of two structural divalent cation sites.

**Structural evidence for the double divalent cation mechanism of Dnase1 family catalysis.** The two leading models of Dnase1-mediated DNA catalysis are the single-cation carboxylic acid and the double divalent cation model[9–11]. The acidic crystallographic conditions of all previous structures precluded corroboration of either proposed model with respect to catalytic divalent cations[10,24,33–38]. Dnase1L3 crystallized with two $Mg^{2+}$ in the active site (Fig. S6), which provides the first structural evidence of the mutagenesis-based double divalent cation model for Dnase1 family catalysis[11] (Fig. 4a, b). To examine the interactions of the

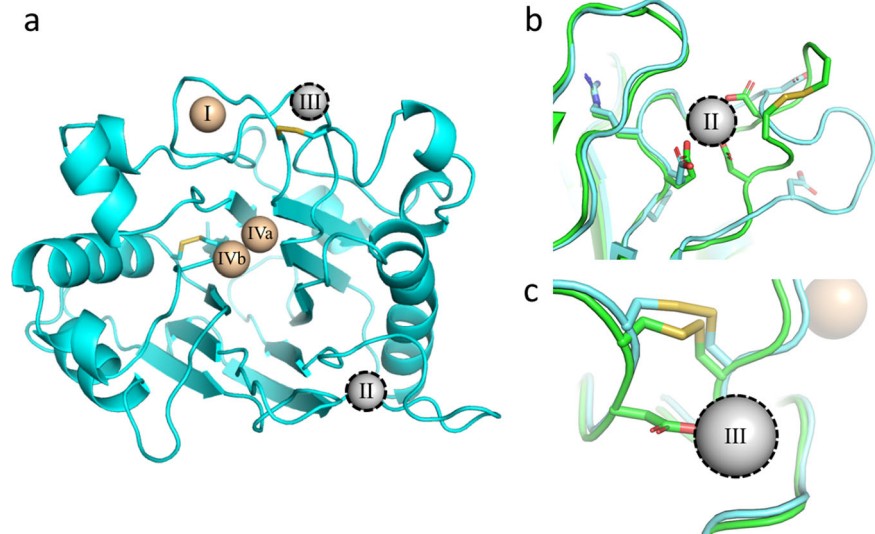

**Fig. 3 Dnase1L3 lacks structural divalent cations present in Dnase1. a** An overview of the divalent cations previously observed in the Dnase1 family superimposed onto the Dnase1L3 structure[10,24]. Metal ions colored tan were observed in the Dnase1L3 crystal structure. The outlined metal ions in grey are absent in Dnase1L3, but previously observed in Dnase1. **b** View of missing structural divalent cation site 'II', observed in previous Dnase1 crystal structures[24]. Dnase1L3 is missing a disulfide bridge that stabilizes the loop structure of site 'II'. **c** View of missing structural divalent cation site 'III', observed in previous Dnase1 (green) crystal structures[24]. The primary coordinating residue in Dnase1, Asp-172, is a Gly-193 in Dnase1L3 (blue).

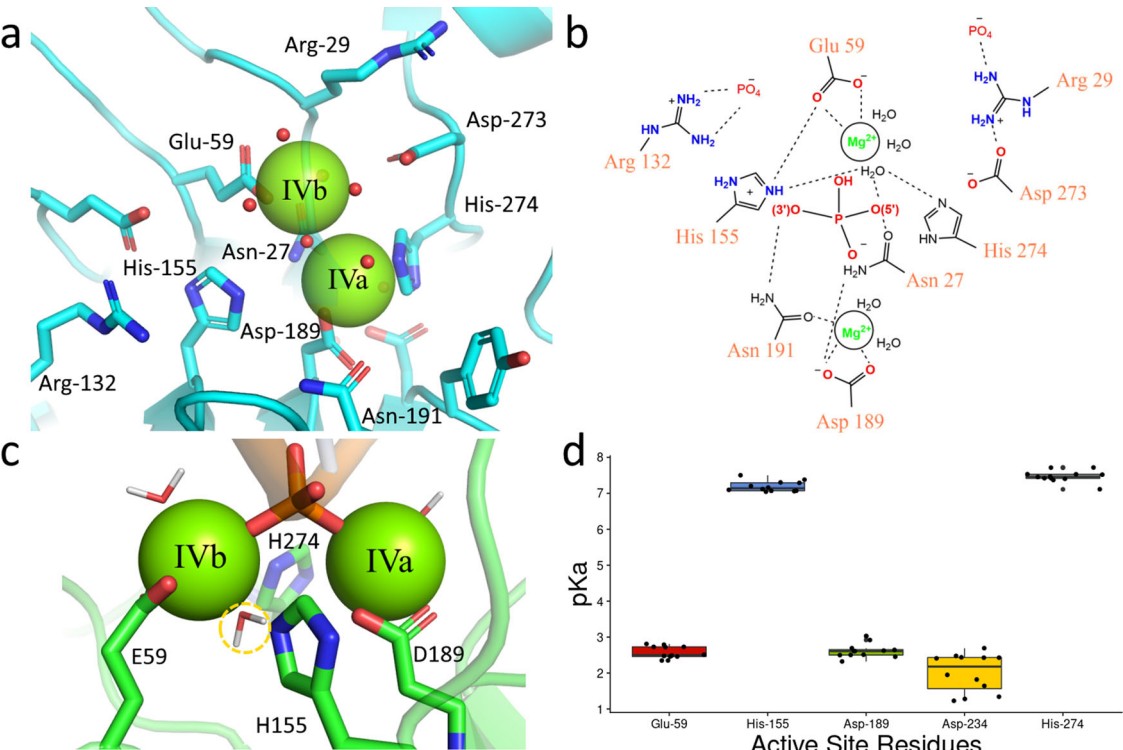

**Fig. 4 The Dnase1L3 catalytic site uses two divalent cations. a** Chain C of Dnase1L3 crystal structure active site showing both divalent cations in the active site. **b** Schematic of the Dnase1L3 active site, based on the crystal structure and molecular dynamics, indicates the roles for key active site residues. **c** Molecular dynamics simulation of Dnase1L3 structure in the presence of dsDNA. A potential deprotonatable water (highlighted in yellow) is coordinated by both His-155 and His274 and positioned for backside attack of the scissile phosphate consistent with the SN2 reaction[71]. The scissile phosphate is coordinated by both divalent cations, and the active site His residues are positioned for acid-base catalysis. **d** The acid dissociation constant for active site residues (with Dnase1L3 numbering) calculated from crystal structures of Dnase1 and Dnase1L3[10,24,33–38]. The acid dissociation constant error bars correspond to calculated standard deviation, $n = 12$ independent models used to calculate the pKa for each residue.

active site in the context of substrate DNA, we used all-atom molecular dynamics. We modeled DNA into the active site of Dnase1L3 based on the structure of Dnase1 crystallized with nucleic acid[33]. In our MD simulations, the two active sites $Mg^{2+}$ ('IVa' and 'IVb') each coordinate the oxygen atoms (OP1 and OP2) of the scissile phosphate of the DNA substrate. We observed a potential deprotonatable water coordinated by the 'IVb' divalent cation (Fig. 4c). Furthermore, His-155 and His-274 are in position for classical acid-base catalysis[11] (Fig. 4c). The catalytic residue proposed in the single cation carboxylic acid model, Asp-189 (Asp-168 Dnase1), coordinates the 'IVa' divalent cation in our crystal structure and simulations (Fig. 4a, b). We tested the potential of each of the active site residues for acid-base catalysis via in silico pH titration of Dnase1L3 and Dnase1. Calculations of the pKa of Asp-189 from the available Dnase1 structures and the four chains of our Dnase1L3 crystal structure indicate that Asp-189 maintains an acidic pKa (Fig. 4d), which predicts that Asp-189 cannot accept a proton during neutral pH acid-base catalysis (Supplementary Data 5). His-155 and His-274 are better candidates for acid-base catalytic residues, as their calculated pKa's are near neutral pH[39] (Fig. 4d). Therefore, we propose Asp-189 is a divalent cation coordinating residue instead of a catalytic base.

The mechanism by which the putative catalytic acid, His-155, is regenerated for its next round of catalysis has not previously been determined. In our model, the regeneration of His-155 can be accomplished by an Arg-132 substrate-sensing switch. In the Dnase1L3 crystal structure, which lacks DNA, Arg-132 interacts with His-155 to decrease the pKa of His-155 favoring the deprotonated form (Fig. S7a). Based on the MD simulation, in the presence of substrate DNA, Arg-132 switches to bind to the DNA backbone; thus permitting His-155 to accept a proton that can then be donated to the 3'O (Fig. S7b). Our Arg-132 switch hypothesis is in line with previous mutagenesis where the alanine substitution of the Arg-132 homolog in Dnase1 is catalytically inactive[19]. To corroborate our results, we performed fluorescence polarization experiments comparing purified Dnase1L3 R132A and Dnase1L3 WT. Consistent with the predicted role of Arg-132 as a substrate sensing catalytic switch, the R132A mutation decreased DNA affinity by a factor of 5, from a $K_D$ of Dnase1L3 WT = 990 nM to a $K_D$ for R132A = 4.95 μM (Fig. S8) (Supplementary Data 6). Therefore, our data suggests that Arg-132 increases catalytic rate by differentially polarizing His-155 along the reaction path.

Another key residue identified by mutagenesis, Asn-191[19], coordinates the 'IVa' divalent cation in the Dnase1L3 crystal structure and MD simulations (Fig. S7c, d). In our MD simulations with DNA, the δ2 amine of Asn-191 is positioned within hydrogen-bonding distance to the 3'O of the scissile phosphate of DNA (Fig. S7d). Asn-191 H-bonding to the 3'O can stabilize the leaving group oxygen for nucleophilic attack by the protonating His-155 (Fig. 4b). We propose a new role for Asn-191 as both a 'IVa' cation coordinating and 3' ester stabilizing residue in accordance with previous mutagenesis results[19]. Our biophysical findings are consistent with the double divalent cation model of Dnase1 catalysis[11].

**The CTD promotes Dnase1L3 specific activity**. The CTD of Dnase1L3 is a 23 amino acid long, highly basic polypeptide unique to Dnase1L3. The CTD has previously been shown to confer increased degradative capacity on lipid complexed DNA[6]. To determine whether the CTD itself possesses inherent nuclease activity, we cloned the CTD of Dnase1L3 onto an unrelated carrier protein, the SH3 domain from the yeast protein, Abp1. We selected the Abp1 SH3 domain because it is well defined by

NMR spectroscopy[40], and it can aid in the overexpression of recombinant CTD polypeptide. We measured the plasmid degradation activity of the purified SH3-CTD fusion protein and the SH3 domain alone and found that neither SH3 nor SH3-CTD degraded DNA (Figure S1b, c). To measure the effectiveness of the CTD in conjunction with the core Dnase1L3 enzyme, we quantitated the plasmid degradation activity of Dnase1, Dnase1L3, and Dnase1L3 ΔCTD. Recombinant full-length and Dnase1L3 ΔCTD had a 10-fold lower $EC_{50}$, and therefore a higher relative activity, compared to standardized, commercially prepared Dnase1 at the same plasmid substrate concentration (Fig. 5a) (Supplementary Data 7). Because ΔCTD and full-length Dnase1L3 have comparable activity, we conclude that the increased naked plasmid degradation activity of Dnase1L3 relative to Dnase1 is not due to the CTD.

In contrast, the CTD from Dnase1L3 is necessary for Dnase1L3-specific activity against complexed DNA. We determined the contribution of the CTD to the degradation of lipid-complexed or antibody-bound DNA by testing full-length Dnase1L3 and Dnase1L3 ΔCTD with two Dnase1L3 assays: the barrier to transfection assay[16] and an immune complex (IC) degradation assay. Full-length Dnase1L3 reduced transfection of HEK cells to a greater degree than Dnase1 or Dnase1L3 ΔCTD (Fig. 5b) (Supplementary Data 8). Next, we tested Dnase1L3-specific activity against a more physiological DNA substrate, pathogenic autoantibody-DNA ICs. To determine the impact of the CTD on disrupting ICs, we measured chromatin IC degradation by ELISA. The ICs were treated with full-length Dnase1L3, Dnase1L3 ΔCTD, or Dnase1. Antibody-bound chromatin was most efficiently degraded by full-length Dnase1L3 (Fig. 5c) (Supplementary Data 9). We measured a 4-fold decrease in the $EC_{50}$ of Dnase1L3 with ICs compared to Dnase1 and Dnase1L3 ΔCTD (Fig. 5d) (Supplementary Data 10). Thus, the CTD of Dnase1L3 mediates the enhanced degradation of DNA in lipid- and antibody complexes, but not naked plasmid DNA.

**The CTD of Dnase1L3 is a disordered DNA binding domain**. The mechanism by which the CTD confers Dnase1L3-specific activity remains unknown. One hypothesis is that the Dnase1L3 CTD forms a C-terminal α-helix that binds and penetrates microparticles and lipids to liberate DNA from DNA-lipid complexes[6]. We tested this hypothesis by incubating full-length Dnase1L3 and Dnase1L3 ΔCTD with pre-defined liposomes (20:20:10:50/PC:PE:PS:cholesterol) and performed a co-sedimentation assay. Surprisingly, Dnase1L3 ΔCTD bound lipids with higher avidity compared to full-length Dnase1L3 (Fig. 5e). Furthermore, neither SH3-CTD nor SH3 alone bound phospholipids (Fig. 5e). Next, we tested whether the CTD bound to microparticles that were purified from HeLa cells. Similar to our results with liposomes, the CTD decreased microparticle interaction (Fig. 5f). Therefore, we conclude that the CTD does not participate in lipid binding.

We next determined the ability of the CTD to interact with DNA. To experimentally determine DNA binding, the CTD was cleaved from the SH3-CTD fusion protein using TEV protease and then purified away from the SH3 and TEV. DNA binding experiments were performed using the purified CTD peptide titrated into fluorescently-labeled 40-mer dsDNA in 100 mM NaCl at pH 7.4. In the fluorescent anisotropy assay, reduced tumbling of the dsDNA indicated CTD binding. The resulting $K_D$ is 4.9 μM at 100 mM NaCl (Fig. 6a) (Supplementary Data 11). Therefore we conclude the CTD is a DNA-binding domain; however, the specific structural interactions between the CTD and DNA are still unknown.

To examine if the CTD possessed discernible secondary structure, we characterized the solution-state structure of the CTD, using the

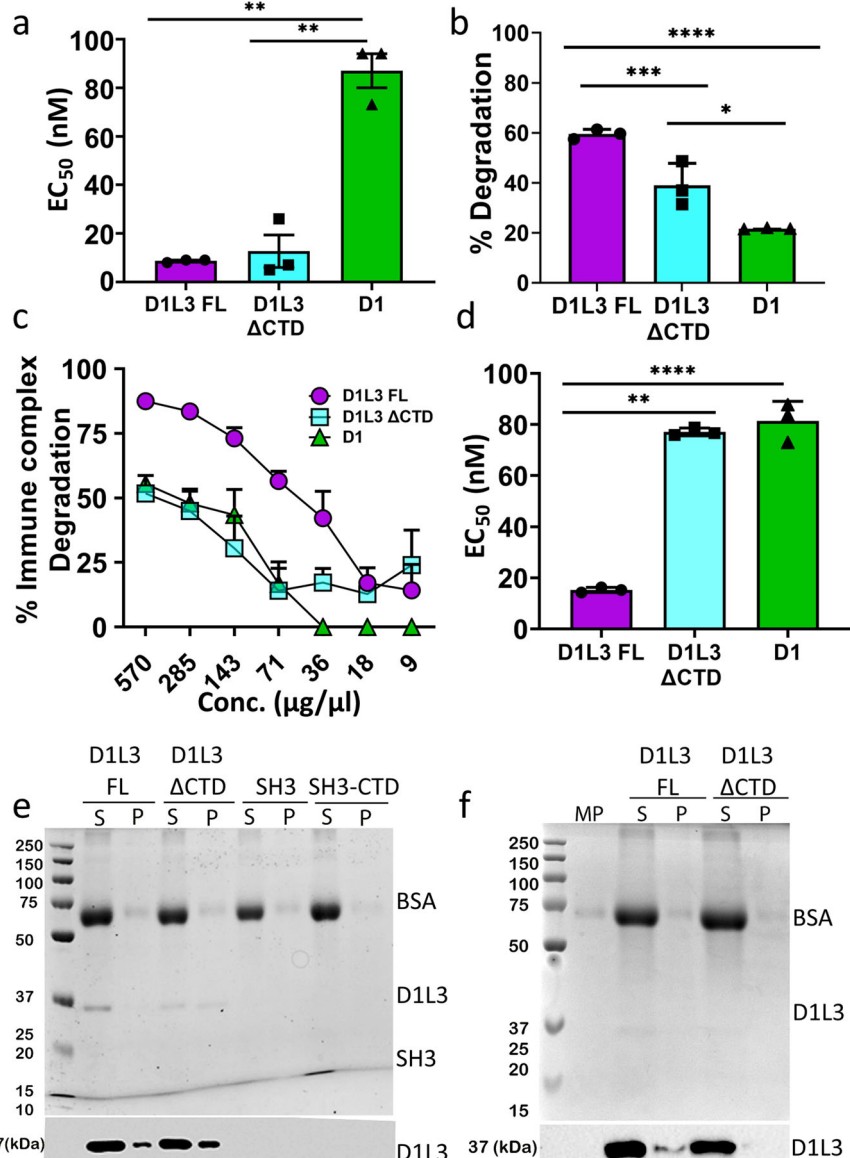

**Fig. 5 The CTD promotes complexed DNA degradation without lipid binding. a** Plasmid degradation activities for full-length Dnase1L3 (D1L3 FL), Dnase1L3 ΔCTD (D1L3 ΔCTD) and Dnase1 (D1) were measured by mixing 200 ng of plasmid DNA with the reported range of Dnase concentrations in a 10 μl reaction volume, for 30 min at 37 °C. **b** Dnase1L3-specific activity was measured using the barrier to transfection assay. HEK cells were transfected with 100 ng of eGFP-N1 plasmid after plasmid-lipid complexes were incubated with 100 ng of the indicated Dnase at 37 °C for 30 min. Transfection efficiency was measured by flow cytometry. **c**, **d** Dnase1L3-specific activity was monitored by measuring immune complex degradation. The indicated concentration of Dnase was incubated with chromatin-anti-dsDNA immune complexes, and the remaining anti-dsDNA antibody was measured. The percent immune complex degradation and EC$_{50}$ were calculated as described in the methods. **e**, **f** The CTD does not promote lipid nor microparticle binding: (**e**) Liposomes or (**f**) microparticles (MP) were incubated with wild type Dnase1L3 (D1L3FL), Dnase1L3 ΔCTD (D1L3 ΔCTD), SH3 or SH3-CTD and the supernatants (S) and pellets (P) were prepared in SDS sample buffer. Samples were resolved by SDS-PAGE followed by Coomassie staining (top) or transferred to nitrocellulose and probed with anti-Dnase1L3 (bottom). Graphs represent mean ± SEM of $n = 3$ independent experiments (**a–d**). Each blot or Coomassie gel is a representative image from four independent experiments. ****$p < 0.0001$, ***$p < 0.005$, **$p < 0.01$, *$p < 0.05$.

SH3-CTD fusion protein, by solution-state NMR spectroscopy. Standard triple-resonance backbone assignment experiments were utilized to assign the whole SH3 domain and linker and 74% of the CTD (Fig. 6b). The NMR data indicate that the CTD exists in a variety of conformations. The backbone resonances attributed to the CTD were used for CS-Rosetta[41] structure determination, which yielded a number of structures containing a small N-terminal helix (Fig. S9a) followed by largely disordered regions (Fig. S9a,b). To corroborate the NMR results, we performed circular dichroism spectroscopy (CD) with synthetic CTD peptide. The resulting CD spectra were consistent with a disordered loop structure (Fig. 6c) in

agreement with our NMR results. There is a possibility that the previously proposed helical CTD model[6] was induced by binding to substrate. To test whether the isolated CTD changed its secondary structure as a function of substrate, we titrated dsDNA into CTD peptide and measured the resulting secondary structure by CD. The addition of substrate did not induce a conformational change in the CTD (Fig. 6c) (Supplementary Data 12). Therefore, we conclude that the CTD is an intrinsically disordered polypeptide even in the presence of DNA substrate.

To further investigate the molecular mechanism of the CTD-DNA interaction, we performed molecular dynamics simulations

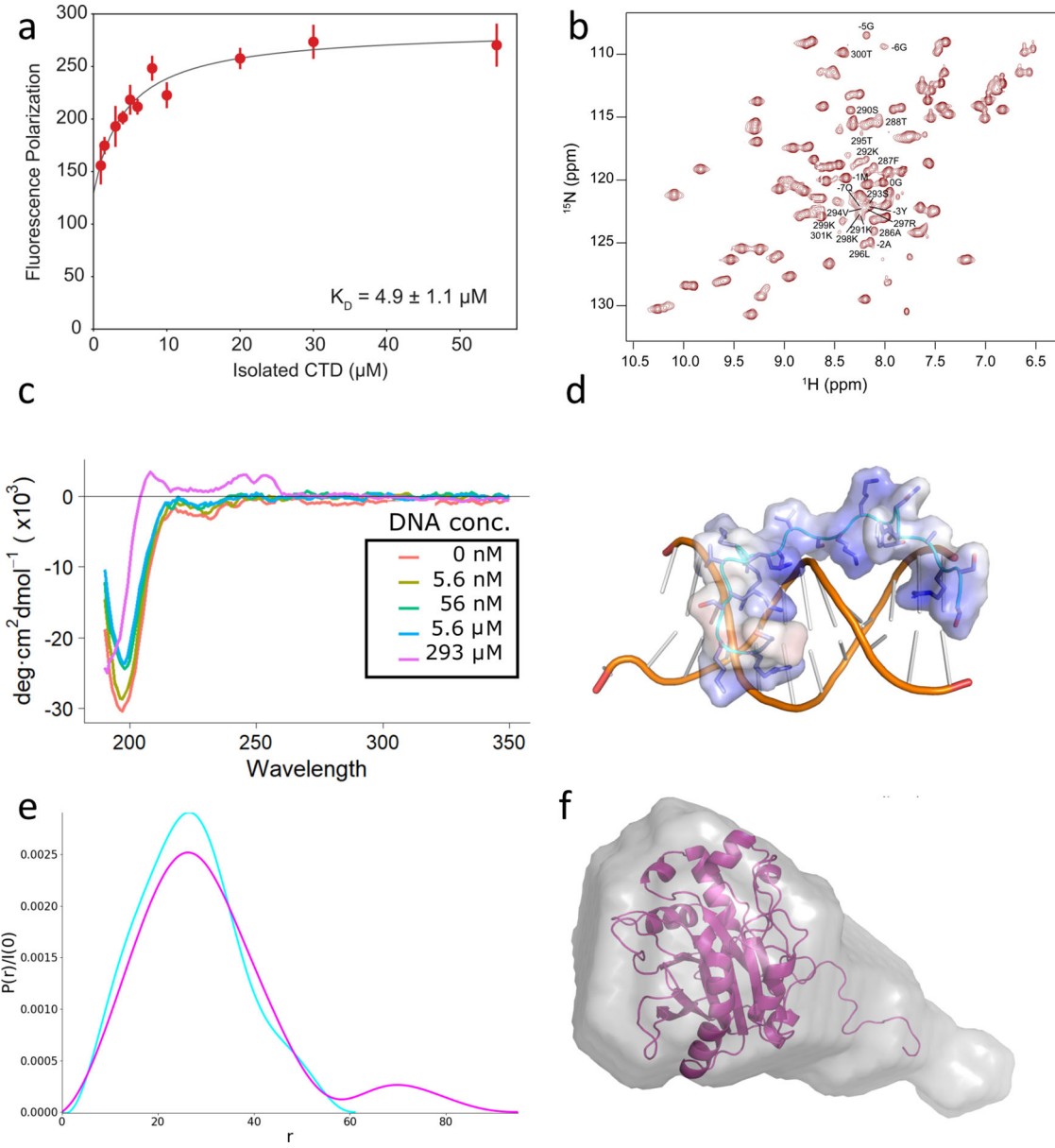

**Fig. 6 The CTD interacts with DNA. a** Fluorescence Polarization of the CTD peptide with a fluorescently labeled 40-mer of dsDNA in 100 mM NaCl. Error bars represent SEM, $n = 42$ independent fluorescence polarization measurements (3 replicates for each concentration). **b** NMR HSQC spectrum of SH3-CTD, which corresponds to the highly ordered and conserved SH3 domain and disordered CTD. The residues of the CTD domain (in Dnase1L3 numbering, missing the last 4 AA's peaks which are unidentifiable) and the linker from the SH3 construct (numbered −7 to 0) are labelled, the SH3 domain constitutes the remaining peaks. **c** Far-UV Circular Dichroism of the CTD peptide with increasing concentration of substrate dsDNA. **d** Snapshot of the CTD peptide interacting with DNA from MD simulation. **e** Pair distribution function created using GNOM as implemented in BioXTAS RAW. The SAXS data indicates a population of scattering atoms ~40 Å from the centroid of Dnase1L3 that was identified as the CTD. **f** SAXS bead model fit, calculated using MultiFOXS, of the combined Dnase1L3 + CTD model.

with a 16-mer of dsDNA and the NMR structure of the CTD equilibrated in an explicit solvent box. The MD simulations were run for a total of 1 μS in NAMD using the CHARMM36m forcefield. Our MD results corroborate the proposed DNA localizing role of the CTD with an immediate attraction towards the dsDNA (Fig. 6d) (Fig. S10a, b). The CTD was flexible during DNA binding and stabilized to two distinct conformations between molecular dynamics simulations (Fig. S10c, d). However, between simulations the intrinsically disordered CTD remained disordered during its interaction with DNA (Fig. 6c). In summary our data shows that the intrinsically disordered C-terminal domain interacts with DNA to increase Dnase1L3's net affinity for its DNA substrate.

**The integration of the core enzyme and C-terminal domain structures.** The structure of full-length Dnase1L3 was solved using SAXS (Small Angle X-ray Scattering) to combine the high-resolution X-ray crystal and the CTD NMR structures. The solution-state SAXS profiles of full-length Dnase1L3 and Dnase1L3 ΔCTD were both collected to describe the CTD's contribution to the full-length enzyme. As expected, recombinant Dnase1L3 ΔCTD is a globular protein in solution, with a radius of gyration ($R_g$) of 21 Å ± 0.4. The full-length Dnase1L3 maintains a similar $R_g$ of 24.1 Å ± 2.5, but the pairwise distribution function indicates a population of scattering atoms roughly 44 Å away from the centroid of the core domain (Fig. 6e) (Supplementary Data 13). The distance between these two populations of

scattering atoms correlates to an almost fully extended CTD separated from the core enzyme. Bead model reconstructions of full-length Dnase1L3 accounted for this extra density with an extended tail volume away from the globular core domain (Fig. 6f). MultiFOXS[42] models generated from the indirect Fourier transform of full-length Dnase1L3 indicate a discrete solvent-exposed CTD (Fig. 6f). The pair distance distribution function obtained from solution-state SAXS analysis shows that full-length Dnase1L3 is a two-domain, pear-shaped molecule[43] (Fig. 6f). Our results indicate that the CTD is an intrinsically disordered polypeptide that is extended from the body of the core enzyme.

## Discussion

Soluble serum nucleases such as Dnase1 and Dnase1L3 digest antigenic nucleic acids that are complexed within discarded membranes or with DNA-binding proteins. Given the complexed nature of the substrate DNA, it is less exposed and therefore less available for digestion by Dnase1. Fortunately, the activity of Dnase1L3 is significantly broader[6,8]. The importance of Dnase1L3 in clearing antigenic DNA is highlighted by the human diseases that result from mutations within the *DNASE1L3* gene. To understand Dnase1L3's broad range of nucleic acid degradation activity, we performed a structure-function analysis to resolve the features of Dnase1L3 responsible for the increased activity. We combined the high-resolution X-ray structure of the core domain with solution-state data of the intrinsically disordered C-terminal domain (CTD) to develop a structure of the complete Dnase1L3 molecule using Dnase1L3 expressed in *E. coli*.

Nucleases are notoriously difficult to overproduce in bacteria due to their DNA laddering activity on the exposed bacterial chromosome[18]. We overcame this hurdle by expressing Dnase1L3 as an enzymatically inactive fusion protein. Dnase1L3 could be switched to an enzymatically active state through proteolytic cleavage of the MBP fusion partner and subsequent purification of the Dnase1L3 enzyme (Fig. S1a). The MBP-Dnase1L3 fusion protein approach allowed us to take advantage of the solubility enhancement from MBP[44], in addition to the potential steric inhibition of Dnase1L3 provided by MBP during translation. We predict that this expression system can be utilized to produce other nucleases besides Dnase1L3.

The enzymatic activity of nucleases such as Dnase1 and Dnase1L3 are particularly important in diseases such as cystic fibrosis (CF). The viscous sputum that occurs within the lungs of CF patients contains DNA and actin[45], both of which contribute to the abnormal visco-elasticity of pulmonary sputum[46]. Utilizing its role as a general nuclease, Dnase1 can act as a mucolytic that digests the DNA component of sputum in the lower respiratory tract[45]. However, the excess of actin within the sputum abrogates the activity of native Dnase1. To address the actin limitation, Dnase1 has been reengineered to overcome actin inhibition (Pulmozyme®); however, in situ actin inhibition remains a therapeutic concern[23]. Dnase1L3 is naturally resistant to actin inhibition, in part, due to Phe-134 that disrupts the 'central hydrophobic interface' with actin in Dnase1[12,25]. A similar mechanism of actin resistance was proposed for Dnase1L2[47]. Other determinants for actin resistance in Dnase1L3 include the disruption of stabilizing salt-bridged networks needed for actin binding to Dnase1, and the electrostatic repulsion of actin at the binding interface. Therefore, Dnase1L3 is naturally resistant to actin inhibition due to predicted and previously undescribed structural features.

The frequency of the R206C single nucleotide polymorphism (SNP) allele of *DNASE1L3* is fairly common (1.7 to 7.7%) in various populations[26]. In fact, R206C homozygosity is causally associated with rheumatoid arthritis[48], systemic scleroderma[49], and systemic lupus erythematosus (SLE)[7]. From a biochemical perspective, the R206C SNP displays decreased or absent[26] enzyme activity and reduced enzyme secretion[27]. Previous literature explained the decreased enzyme activity in terms of increased mobility of the loop preceding Arg-206 (Dnase1L3 residues 193–196) that disrupts nucleic acid binding[28]. In addition, the authors concluded that only the interaction between Arg-206 and Glu-170 is relevant to the hypofunctional phenotype[28]. However, our crystallographic and molecular dynamics results reveal a more extensively salt-bridged network between Arg-206 and Asp-166, Glu-167 and Glu-170. Further, ANS fluorescence experiments demonstrate that the R206C mutation disrupts protein folding. Our MD analysis indicates there is a general increase in the calculated free energy of folding as a result of the loss of the critical Arg-206 salt bridging interactions (Fig. S4a). Indeed, we find that the R206C mutation alters the calculated protein stability by 14 kcal/mol, consistent with three salt bridges based on the crystal structure (Fig. 2a, c). It therefore follows that errant protein folding would also affect extracellular secretion of the enzyme[27] that results in a net hypofunctional Dnase1L3.

In our crystal structure and molecular dynamics simulations of Dnase1L3, Asn-191 coordinates $Mg^{2+}$ 'IVa' consistent with previous mutagenesis results[19]. Our analysis of the Dnase1L3 active site revealed a potential mechanism of His-155 regeneration via Arg-132, consistent with mutagenesis data and the proposed double divalent cation model[11]. The principal dissenting Dnase1 catalytic mechanism is the single-cation carboxylic acid model, which derives from the 4AWN crystal structure[10]. We show that Asp-189 in Dnase1L3 (homologous to Asp-168 in Dnase1) instead binds to the previously predicted second active site divalent cation, 'IVa'[32]. The disagreement in $Mg^{2+}$ coordination via Asp-168 (Asp-189 in Dnase1L3) can be explained by the more acidic pH of crystallization of 4AWN[10]. We provide the first structural evidence to support the double divalent cation model for the Dnase1 family[11]. Our structural evidence clarifies the number of active site divalent cations and the role of several catalytic residues in the Dnase1 family. Further investigation is needed to resolve the means of electrophilic attack and protonation of the 3'O on the scissile phosphate.

Aside from catalytic divalent cations, Dnase1L3 binds structural divalent cations differently than previously observed in Dnase1. The two missing divalent cation binding sites in Dnase1L3 have been described as structurally stabilizing binding sites in Dnase1[32]. Furthermore, the disruption of the 'II' and 'III' binding sites in Dnase1 decreases the measured enzyme activity[50]. Nevertheless, in the absence of these stabilizing cations, the conformations of the catalytic residues in Dnase1L3 are similar to those observed in Dnase1. To address the perceived lack of stabilizing cations in Dnase1L3, we propose that the 'II' and 'III' sites observed in Dnase1 are unnecessary due in part to additional disulfide-mediated stabilization. The missing 'III' cation binding site is proposed to stabilize: the active site, the disulfide bridge, and nearby DNA-binding residues in Dnase1[50]. In Dnase1L3, the absence of the Dnase1 'III' coordinating residue Asp-172 (Gly-193 in Dnase1L3) does not disrupt the disulfide located between Cys-194 to Cys-231. The additional disulfide bond present in Dnase1L3 but not Dnase1, Cys-24 to Cys-52, may allow for the reduced necessity of divalent cation-mediated stabilization.

Dnase1L3 displays an enhanced substrate diversity through its unique, highly basic C-terminal domain (CTD)[5,6]. Given the physiological role of Dnase1L3 and the exaggerated charge of its CTD, two binding targets were proposed: phospholipid membranes and DNA. Our biophysical characterization of Dnase1L3 reveals that the CTD is an extended, intrinsically disordered

domain that binds dsDNA with μM affinity at physiological salt concentrations. Dnase1L3 showed no lipid or microparticle partitioning, signifying the CTD is indeed a DNA affinity domain and not a lipid binding domain. Functionally, the intrinsically disordered CTD possesses the conformational versatility to bind complexed DNA in various forms, including nucleosomal, microparticle, and immune complexed DNA. We propose that the CTD is a conformationally flexible, disordered domain that enhances heterogeneous DNA binding and confers the unique activity attributed to Dnase1L3.

## Methods

**Cloning and expression**. The genes for full-length human Dnase1L3 and the Dnase1L3 ΔCTD were sub-cloned into BamHI/XhoI sites in a Tobacco-Etch Protease (TEV) cleavable, his-tagged, maltose-binding protein (MBP) expression vector for *E. coli* expression. The expression plasmid was transformed into Rosetta-Gami®2 (DE3) competent cells (Novagen) to maintain disulfide bond formation. Single colonies were picked and grown to confluence in 100 ml of LB media. 10 ml of culture was inoculated into 1 L of Terrific Broth (TB) media, and grown to $OD_{600}$ of 2.0. The culture was induced with 400 μM IPTG and grown at 18 °C for 18 h. Cells were harvested by centrifugation at 18,000 rpm in an SS-40 rotor for 20 min. Cell pellets were frozen at −80 °C until used for purification.

SH3-CTD was expressed in BL-21-Gold (DE3) *E. coli* cells (Agilent) in 3 L of Terrific Broth, induced with 400 μM IPTG and grown at 18 °C for 18 h. Cells were harvested by centrifugation, frozen in liquid $N_2$ and stored at −80 °C until ready for use.

**Purification of full-length Dnase1L3, Dnase1L3ΔCTD, SH3-CTD and Dnase1L3-MBP fusion**. 15–20 g of cells were resuspended in 160 ml of lysis buffer (20 mM HEPES pH 7.4, 300 mM NaCl, 1 mM $CaCl_2$). The cells were lysed using a microfluidizer and spun down at 19,500 rpm in a JA20 rotor for 50 min. The supernatant was collected and run through a newly regenerated $Ni^{2+}$/NTA column. The column was washed with lysis buffer that contained 30 mM imidazole. The protein was eluted with lysis buffer that contains 250 mM imidazole, 5 mM maltose. 0.1 mM PMSF was added to the eluted sample. To remove the MBP fusion protein and to isolate the Dnase1L3, 2–3 mg of TEV protease were added to the sample and incubated at 4 °C overnight. An SDS gel was run to ensure the fusion protein was cut by TEV before moving on to the next purification step. The sample was diluted to 100 mM NaCl with Buffer A (20 mM Hepes pH 7.4, 1 mM $CaCl_2$). An SP-Sepharose (cation exchange) column was prepared and the sample was run through the column while collecting the flow-through (which contains the MBP). The column was then washed with Buffer A until the UV returned to the baseline before the sample was loaded onto the column. The protein was eluted with a gradient of 0–100% Buffer B (20 mM Hepes pH 7.4, 1 M NaCl, 1 mM $CaCl_2$) with a volume of at least three times the bed volume of the column. Purity was measured by SDS-PAGE. Finally, the sample was concentrated to run on an Superdex-75 gel filtration column equilibrated with Buffer C (20 mM HEPES pH 7.4, 400 mM NaCl, 1 mM $CaCl_2$). The peak was then run on an SDS gel and imaged using a BioRad Stain-Free Gel imaging system.

To purify SH3-CTD, centrifuged bacteria were lysed by microfluidization and insoluble cell components were pelleted with 19,500 x g centrifugation. SH3-CTD was separated from soluble lysate with 6-His $Ni^{2+}$/NTA affinity chromatography. The high-affinity fraction containing SH3-CTD was eluted at 250 mM Imidazole concentration. The elution was concentrated to 1 ml and injected into a 75 ml Superdex-75 size exclusion column. The monodisperse fraction was exchanged into 1 M NaCl and incubated overnight with 1 mg of TEV. The CTD was separated from the SH3 and TEV using Ni-NTA affinity chromatography, the CTD fraction came out in the FT. The CTD was buffer exchanged using 700 Da cutoff filters into 100 mM NaCl and concentrated with lyophilization.

The wild-type Dnase1L3 and Dnase1L3 R206C MBP fusion proteins were produced and purified in a similar manner as the full-length Dnase1L3 purification methods described above. However, after $Ni^{2+}$/NTA affinity chromatography, no TEV protease was added to the sample. Instead the fusion protein was purified to homogeneity by Q-Sepharose anion exchange chromatography followed by size exclusion chromatography. At each step, the purity was assessed by SDS-PAGE.

**Cell culture**. HeLa cells (ATCC (Manassas, VA, USA), CCL-2) and HEK cells (ATCC CRL-1573) were cultured at 37 °C and 5% $CO_2$ in DMEM (Corning, Corning, NY, USA) supplemented with 10% Equafetal bovine serum (Atlas, Fort Collins, CO), 1 × L-glutamine (D10) and 1 × penicillin/streptomycin. All cell lines were negative for mycoplasma.

**Antibodies**. The anti-Dnase1L3 rabbit polyclonal antibody was obtained from Abnova, and the anti-dsDNA monoclonal antibody was from the Developmental Studies Hybridoma Bank (DSHB, Iowa City, IA). Clone autoanti-dsDNA was deposited with the DSHB by Voss, E.W. (DSHB Hybridoma Product autoanti-

dsDNA). HRP-conjugated secondary antibodies were from Jackson Immunoresearch (West Grove, Pennsylvania).

**Plasmid degradation assay**. Plasmid degradation assays were performed in a 10 μl reaction volume, 200 ng of plasmid DNA was incubated with varying concentrations of Dnase1L3 full length, Dnase1L3 ΔCTD, Dnase1, MBP-Dnase1L3, SH3-CTD or SH3 alone in Dnase assay buffer (20 mM Tris, pH 7.4, 5 mM $MgCl_2$, 2 mM $CaCl_2$) for 30 min at 37 °C[51]. The extent of DNA degradation was quantitated by measuring the integrated intensity of degraded and intact plasmid DNA from Gel Red-stained agarose gels using Photoshop Creative Suite (Adobe, San Jose, CA) and determining the percent degradation. The $EC_{50}$ for plasmid degradation was calculated from the dose-response curve using logistic regression.

**Barrier to transfection**. HEK cells (ATCC CRL-1573) were plated 1 day prior to the assay at $5 \times 10^5$ cells per well in a 24 well plate. DNA-lipid complexes were prepared by incubating 100 ng of eGFP-N1 plasmid (Takara Biosciences) with Lipofectamine 2000 for 20 min. DNA-lipid complexes were then incubated with 100 ng each of full-length Dnase1L3, Dnase1L3 ΔCTD or Dnase1 at 37 °C, 5% $CO_2$ for 30 min. HEK cells were then transfected with the control or the Dnase treated DNA-lipid complexes and incubated for 48 h. The cells were supplemented with fresh D10 media after 24 h. Cells were harvested, washed in FACS buffer (2% fetal calf serum, 0.05% $NaN_3$ in 1x PBS), and analyzed on an Attune NxT flow cytometer. Transfection efficiency was 70% in control cells. Reduced transfection efficiency was calculated compared to control transfected cells.

**Immune complex degradation**. To measure immune complex degradation, a modified ELISA protocol was used. ELISA plates were precoated with 0.05 mg/mL poly-L-lysine at room temp for 20 min, washed with 1x nuclease-free water, and coated with 5 μg/ml calf thymus DNA (Sigma) overnight at 4 °C. After washing 3 x in PBS with 0.05% Tween (PBST) and blocking for 1 h at room temp with 1% BSA in PBST, 250 pg/mL anti-dsDNA antibody was added to all wells except the standard curve. The standard curve received 2-fold dilutions of anti-dsDNA antibody at 500 pg/mL. After 1 h, the plates were washed 3x in PBST, Dnase (diluted into 20 mM HEPES, pH 7.4, 300 mM NaCl, 1 mM $CaCl_2$) was added, and plates incubated 37 °C for 2 h. Plates were washed 3x in PBST, incubated with HRP conjugated goat anti-mouse IgG antibody (1:20,000), and developed using 0.2 mg/ml TMB (Sigma), 0.015% $H_2O_2$ in 100 mM sodium acetate, pH 5.5. The reaction was stopped with 0.5 M $H_2SO_4$. $A_{450}$ was measured and antibody concentration in each well calculated. The percentage of dsDNA antibody remaining in the well was calculated by comparison to control. Percentage immune complex degradation was 100 - %remaining antibody. The $EC_{50}$ for each Dnase was calculated using logistic regression.

**Generation of microparticles**. Microparticles from HeLa cells (ATCC CCL-2) were generated by treatment with staurosporine to make the cells apoptotic[52]. Cells were cultured overnight and then treated with 1 mM staurosporine (Sigma-Aldrich) for 8 h in serum-free DMEM. Microparticles were harvested and debris removed by centrifugation for 5 min at 1500 rpm. The supernatant was collected and centrifuged at 20,000 x g for 30 min to pellet the microparticles.

**Generation of liposomes**. Synthetic liposomes were prepared by mixing 20:20:10:50 mol percent of phosphatidylcholine (Avanti), phosphatidylethanolamine (Avanti), phosphatidylserine (Avanti) and cholesterol (Sigma). This lipid mixture was dried under $N_2$ and resuspended to a final concentration of 4 mg/mL in liposome buffer (15 mM HEPES pH 7.2, 1 mM $Mg(CH_3COO)_2$ and 50 mM sorbitol). Liposomes were incubated at 37 °C for 1 h, freeze/thawed using dry ice/37 °C bath 5 times and stored at −80 °C until further use.

**Lipid and microparticles binding assay**. Microparticles prepared from HeLa cells (ATCC CCL-2) or 4 mg/ml liposomes were incubated with 10 mg/mL bovine serum albumin carrier protein and 6 μM full length Dnase1L3, Dnase1L3 ΔCTD or Dnase1 in 1x PBS with $Ca^{2+}$ and $Mg^{2+}$ on ice for 1 h. Microparticles or liposomes were then centrifuged at 20,000 x g 4 °C for 15 min and the supernatant saved for SDS-PAGE analysis. The pellet was washed with 1x PBS with $Ca^{2+}$ and $Mg^{2+}$ and centrifuged at 20,000 x g 4 °C for 15 min. The resulting pellet was resuspended in SDS sample buffer. Samples were resolved on a 12.5% gel and either Coomassie-stained or transferred to a 0.45-μm nitrocellulose membrane. Immunoblotting with anti-Dnase1L3 (1:1000) was followed by anti-rabbit IgG conjugated to HRP (1:10,000). Antibody staining was visualized on a FluorChemE (Protein Simple, San Jose, CA, USA) using enhanced chemiluminescence reagent [0.01% $H_2O_2$ (Walmart, Bentonville, AR, USA), 0.2 mM p-Coumaric acid (Sigma-Aldrich), 1.25 mM Luminol (Sigma-Aldrich), and 0.1 M Tris, pH 8.4]. Immunoblots were analyzed using Photoshop Creative Suite 3.

**Fluorescence polarization**. The dsDNA binding assays with the CTD were performed using a FAM (carboxyfluorescein)-labeled 40-mer DNA (5′-FAM-GTGTTCGGACTCTGCCTCAAGACGGTAGTCAACGTGCTTG-3′ and 5′-CAAGCACGTTGACTACCGTCTTGAGGCAGAGTCCGAACAC-3′, IDT)[53]. The

fluorescence polarization binding assays for Dnase1, Dnase1L3 WT and Dnase1L3 R132A were performed using the same 40mer sequence labelled with FITC (fluorescein 5(6)-isothiocyanate). The dsDNA was made by annealing the fluorophore-labeled ssDNA to an unlabeled complementary strand at 95 °C. Fluorescence polarization (FP) experiments for the CTD were performed in binding buffer (50 mM Hepes pH 7.5, 100 mM NaCl, 0.1 mM EDTA, 1%Glycerol, 1 mM TCEP, 0.1% PEG 6000 w/v), with 20 nM dsDNA, and an increasing amount of isolated CTD for a final volume of 25 µL. FP experiments for Dnase1L3 R132A were performed similarly, but the binding buffer contained increased EDTA, 1 mM to inhibit DNA degradation. Samples were incubated for 30 min at 37 °C in a 384-well black plate (Fisher Scientific) before reading for FAM or FITC-FP in a Synergy Neo2 plate reader using the FP 485/530 filter. Three technical replicates were performed. The apparent dissociation constants were calculated by fitting the data to a modified version of the Hill equation[54].

**ANS fluorescence**. Each of the protein samples were diluted into 20 mM HEPES, 400 mM NaCl, 1 mM CaCl$_2$ pH 7.4 to a concentration of 2.2 µM. The protein was incubated for 30 min at 25 °C with 25 µM ANS (8-anilino-1-naphthalenesulfonic acid). ANS was excited at 365 nm and fluorescence was recorded from 400 to 600 nm for buffer, the MBP control sample, Dnase1L3 WT fusion protein, and Dnase1L3 R206C fusion protein in a 96 well plate using the BioTek Synergy4 Multimode Plate Reader. The excitation/emission slit width was set to 5 nm. The mean fluorescence for each wavelength was taken from three independent measurements, and the mean ± standard deviation is reported for each datapoint.

**Crystallization, data collection, structure solution and refinement**. Purified Dnase1L3 ΔCTD was concentrated to 8 mg/ml and was screened with multiple sparse matrix crystallization kits. Initial crystal hits were expanded to improve crystal quality and yield more useful crystals for X-ray analysis. The final crystallization condition was 19% PEG 8000, 250 mM MgCl$_2$, 100 mM Tris, pH 8.5 at 10 °C. The final Dnase1L3 ΔCTD crystals grew to a final size of ~0.5 mm in 60 days and were obelisk-shaped. The Dnase1L3 ΔCTD crystallized in the P1 space group with four molecules in the asymmetric unit (Table 1). The final X-ray dataset was collected at the SLAC beamline 14-1. The Dnase1L3 ΔCTD crystals diffracted to 1.9 Å .

The structure was solved with molecular replacement, as implemented in Phenix[55], using the 1DNK structure as the target. X-ray data to 2.2 Å was used for refinement. Electron density was fit using COOT[56] prior to additional rounds of structural refinement in Phenix. Each of the four molecules in the asymmetric unit was refined independently using Phenix[55].

**Circular dichroism**. A commercially synthesized peptide corresponding to the last 23 amino acids of Dnase1L3 (i.e., C-terminal domain (SSRAFTNSKKSVTLRKKTKSKRS) (United BioSystem Inc, Herndon, VA, US) was resuspended from powder in milli-Q water to a stock concentration of 50 µM. The stock was diluted to 20 nM in 5 mM NaCl with increasing concentrations of DNA from 0 nM to 293 µM of a self-complementary DNA substrate (GCGATCGCGCGATCGC). The circular dichroism data were collected on a Jasco J-815 CD spectrophotometer from 190 nm to 350 nm. Buffer subtractions were the equivalent concentration of oligomer DNA in 5 mM NaCl. The spectra were recorded in CD units of mdeg and then converted to molar ellipticity. The CTD was disordered throughout the measurements; however, the background subtraction became increasingly different from the Sample CD at high concentrations of DNA (293 µM) likely corresponding to differences in the secondary structure of the DNA, even though the peptide remained disordered.

**NMR**. For NMR experiments, transformed Rosetta BL21 cells were grown in 2x minimal M9 media using$^{15}$NH$_4$Cl (1 g/L) and unlabeled or$^{13}$C D-glucose (3 g/L) as sole nitrogen and carbon sources, respectively. Protein expression was induced with 0.5 mM isopropyl-$\beta$-d-thiogalactopyranoside for 18 h at 25 °C. Bacterial cells were suspended in SH3 lysis buffer (20 mM HEPES pH 7.4, 300 mM NaCl). Cell suspensions were lysed in a microfluidizer. Lysates were clarified by centrifugation at 19,500 rpm in a JA-20 rotor for 45 min at 4 °C and loaded on Ni-NTA resin pre-equilibrated with lysis buffer. SH3-CTD constructs were eluted in 250 mM imidazole lysis buffer. The purification was followed by size exclusion chromatography in SH3 lysis buffer. SH3-CTD constructs were then buffer exchanged into 20 mM Hepes pH 7.4, 300 mM NaCl, and 10% D$_2$O and concentrated to a desirable concentration (0.7 mM) for NMR spectroscopy using a 5 kDa molecular weight cutoff centrifugal concentrator (Millipore). All NMR experiments were performed on an Agilent 600 MHz (14.1 T) DD2 NMR spectrometer equipped with a room temperature HCN z-axis gradient probe. NMR data were processed with NMRPipe/NMRDraw[57] and analyzed with CCPN Analysis[58]. Backbone $^{13}$Cα,$^{13}$Cβ,$^{13}$C′,$^{15}$N, and $^1$HN resonance assignments of SH3-CTD were obtained from standard gradient-selected triple-resonance HNCACB, HN(CO)CACB, HNCO, HN(CA)CO[59], HCCH-TOCSY[60], and NOESY HSQC[61] experiments at 22 °C. Assignment data were collected with a random nonuniform sampling (NUS) scheme and reconstruction of NUS spectra was performed using Sparse Multidimensional Iterative Lineshape-Enhanced (SMILE) program[62]. The CTD and the linker between CTD and SH3 domains assignment were isolated by excluding the

SH3 domain assigned residues using an assignment reference from the BMRB (code = 18054)[40].

**SAXS data collection and processing**. SAXS data were collected on a Xenocs BioXolver configured for SAXS/WAXS/GISAXS with a Genix 3D Cu HFVL source and a DECTRIS EIGER 1M detector. Purified full-length Dnase1L3 and ΔCTD were concentrated to 4 mg/ml. The enzyme was monodisperse, as determined by size exclusion chromatography, and was in 400 mM NaCl, 20 mM HEPES, 1 mM CaCl$_2$. The filter concentrator flowthrough buffer was used for SAXS buffer subtraction. The proteins were irradiated five times for 300 s each and averaged before buffer subtraction with BioXtas-RAW. The Indirect Fourier Transform was performed using GNOM[63] with the limits based on the GNOM analysis of IFT quality. Bead models were constructed using Dammif/n[64]. MultiFoXS[42] was used in conjunction with the pairwise distribution function to refine the crystal structure of Dnase1L3 ΔCTD grafted with the CTD structure from NMR using comparative structure modeling[65].

**Statistics and reproducibility**. Prism 5.0 (GraphPad, La Jolla, CA, USA) or Excel were used for statistical analysis. Data are represented as mean ± SEM or standard deviation as indicated. The EC$_{50}$ was calculated by logistic regression. Statistical significance was determined by one-way ANOVA or repeated measures ANOVA; $p < 0.05$ was statistically significant. Graphs were generated in R ggplot2, Excel and Photoshop.

**Molecular dynamics**. All molecular dynamics input files were generated with CHARMM-GUI for NAMD with the CHARMM36m force field[66]. The high-resolution crystal structure of Dnase1L3 was used throughout for MD. The total charge of the system was neutralized by randomly substituting water molecules with Mg$^{2+}$ and Cl$^-$ to obtain neutrality with 0.15 M salt concentration. The TIP3 model for water was used throughout. A switching function was applied to the van der Waal's potential energy from 10 to 12 Å to calculate nonbonded interactions. The Particle Mesh Ewald (PME) algorithm was used to calculate electrostatic interactions. Equilibration runs used the NVT ensemble at 300 K. Energy minimization was performed for 10,000 steps to avoid any bad contacts generated while solvating the system. C-α restraints were generated from VMD. The simulations were analyzed using VMD. Production runs were performed on the HPCC Nocona cluster at Texas Tech University.

The structure for the CTD, as determined by solution-state NMR, was isolated from the SH3 domain along with DNA based on the 1BNA 12-mer of dsDNA. The solution box was built with the CTD and a 12-mer (5′-CGCGAATTCGCG-3′) dsDNA molecule that was placed ~30 Å and 50 Å from the CTD. The distances were required to minimize the bias of the simulation. The simulation was run for a total of 1 µs with neutralizing KCl. VMD and the Bio3D library for R[67] were used to analyze the trajectories.

DNA was added to the system based on a superposition of the 1DNK crystal structure of Dnase1 with DNA. Chain C of the Dnase1L3 high-resolution crystal structure was the starting structure for the MD runs with DNA. A 75Å x 75Å x 75Å water box with 150 mM MgCl$_2$ (charge neutralized) was sized to the input Dnase1L3 molecule and was built using periodic boundary conditions. The simulated system contained ~35,000 solvent atoms, ~69 counter ions, and 4336 protein atoms. The crystallographic Mg$^{2+}$ were retained. Three simulations were run for 1 µs each, starting from separate equilibration runs.

The R206C mutation was generated with PyRosetta using Chain A of the crystal structure and Rosetta relaxed 3 times with the lowest energy score chosen prior to MD simulation. The solution box was built with 150 mM MgCl$_2$ and charge neutralization. The crystallographic Mg$^{2+}$ were retained. A total of three production runs of 500 ns for Arg-206 and Cys-206 were generated for analysis. The last 100 ns (400–500 ns) were sufficient to reach a stable snapshot of each repeat. Ten frames, each 10 ns apart from the last 100 ns of the MD simulations were analyzed for free energy of folding using Rosetta, with the lowest energy score, REF-2015, after three full Rosetta relax cycles compared between Arg-206 and Cys-206. A one-way ANOVA, (alpha = 0.05) was performed to highlight the difference between the single comparison of significance, Arg-206 or Cys-206 on free energy of folding. The average computed free energy of folding from 10 timestamps across 3 simulations, for both WT and R206C Dnase1L3, was measured as −664.99 ($n = 30$) for WT (Arg206) and −650.96 ($n = 30$) for R206C. The standard deviation for WT free energy of folding was 13.633, and 15.642 for R206C. The difference in folding was determined to be 14.03 ± 3.79 (s.d.), $p = 0.000476$.

**Iso-electric point calculations**. Iso-electric calculations were computed using H++[68–70]. For the actin-binding loop following the actin-binding helix in Dnase1[24], and the homologous loop in Dnase1L3, the regions of interest were isolated in Pymol and processed using the H++ web server at a pH of 7, with a salinity of 0.15 M, internal dielectric of 0.1, and external dielectric of 80.

**pKa of active site residues**. The computational determination of acid dissociation constant was done with a local installation of DelPhi-Pka[38]. All four Dnase1L3 molecules in the asymmetric unit and each available Dnase1 structure[10,24,33–37],

were processed for pKa calculation in DelPhi-Pka after being cleaned of any hetero-atoms.

**Reporting summary**. Further information on research design is available in the Nature Research Reporting Summary linked to this article.

## Data availability

All crystallographic coordinates and structure factors have been deposited in the PDB under the accession code 7KIU. The NMR chemical shifts and assignments for SH3-CTD are deposited to the BMRB with accession code 51384. All source data used to generate the graphs and charts in the manuscript is included in the Supplementary Data. All uncropped and unedited blots are included as Figs. S11 through S14.

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

## Acknowledgements

This publication was supported by the Lupus Research Alliance, The CH Foundation, the Lubbock Lupus Group, and the National Institute of General Medical Sciences (1R35GM128906; MPL) and a seed grant provided by the TTUHSC Center for Membrane Protein Research. Use of the Stanford Synchrotron Radiation Lightsource, SLAC National Accelerator Laboratory, is supported by the U.S. Department of Energy, Office of Science, Office of Basic Energy Sciences under Contract No. DE-AC02-76SF00515. The SSRL Structural Molecular Biology Program is supported by the DOE Office of Biological and Environmental Research, and by the National Institutes of Health, National Institute of General Medical Sciences (P30GM133894). The contents of this publication are solely the responsibility of the authors and do not necessarily represent the official views of funding agencies, including the LRA, NIGMS or NIH. The authors acknowledge the Texas Tech University College of Arts and Sciences Microscopy for flow cytometry and the High-Performance Computing Center (HPCC) at Texas Tech University for providing computational resources that have contributed to the research results reported within this paper. http://www.hpcc.ttu.edu The authors also thank Dr. Faraz Harsini for help with the initial Dnase1L3 purification.

## Author contributions

P.A.K., R.B.S., and M.P.L. conceived the experiment(s), J.J.M., M.E., E.M., J.V., B.M. conducted the experiment(s), R.B.S., P.A.K., and M.P.L. analyzed the results. All authors reviewed the manuscript.

## Competing interests

Authors P.A.K., R.B.S., and J.J.M. are inventors on pending TTUS patent applications (US17763913, EP20874456.5) relating to certain research described in this article. The remaining authors declare no competing interests.
