## [Peer Review File · Communications Biology]

Reviewers' comments:

Reviewer #1 (Remarks to the Author):

The manuscript by McCord et al. describes the activity of the C-terminal domain (CTD) of Dnase 1L3 for its function. The explore the function-structure relationships with a suite of experiments including X-ray crystallization, CD and NMR, as well as computational techniques.

While the manuscript appears interesting for me and the results promise some interesting new insights into Dnase activity and its relation to autoimmunity, I found some issues that need to be clarified before considering this manuscript for publication in Communications Biology.

1. The authors show an HSQC spectrum of the SH3 CTD and state that the spectrum provides information on structural variety. I see neither a discussion of nor a proof for that claim. This needs to be clarified.
2. The authors state that the CTD is an IDR, however, the NMR spectrum shows a chemical shift dispersion of over 3 ppm along the proton dimension. How do the authors explain this discrepancy?
3. The MD simulations are said to start from the NMR structure of the CTD calculated by CS-Rosetta. There is no information the structure refinement, e.g., how many NOE constraints were used, or only chemical shift data? How reliable is the model? #
4. If the CTD is an IDP or adopts various structures, how can one refine a reliable NMR structure of it? The authors need to clarify how to obtain this structure.
5. Also they might want to calculate SSP scores from the ^{13}C chemical shifts in their HNCACB experiment to underline the claim that the protein adopts an intrinsically disordered structure.
6. I would also show or deposit the NMR data somewhere, as most groups do by now.
7. The main text states that the MD simulation ran 1 microsecond (also Figure S9), but the data in Figure S5 shows only 500 ns. Why?
8. Continuing with Figure S5: The differences in binding energy between the WT and the R206C mutant are on average 14 kcal/mol. This corresponds to ca. 1/3 of a hydrogen bond. How significant is that? The scatter/box plots in panel B don't look very significant to me, despite a calculated p-value of $5e-4$. Maybe the authors could comment on how they calculate this value?
9. In Figure S9: I don't understand why the trajectory in panels A and B are traced for 12 ns, while the final structures are shown after 1 us? From Figure S5, it appears that the first ca. 20 ns are still subject to very strong structural fluctuations, since the system isn't yet energetically equilibrated.
10. Relating to the above I couldn't find any methods details on the MD simulations. (equilibration, solvent conditions, cut-off length etc.)

Once these points have been addressed, I'm happy to reevaluate the manuscript.

Reviewer #2 (Remarks to the Author):

Dnase1 and Dnase1L3 (Dnase1-like protein 3) are homologous to each other. Although both Dnase1 and Dnase1L3 can degrade DNA in sequence-nonspecific manner, Dnase1L3 plays more critical role in clearing antigenic DNA from serum and Dnase1 could not rescue the phenotype associated with Dnase1L3 deficiencies. In this work, McCord and coworkers performed excellent structural studies of Dnase1L3, solved the core domain structure by X-ray crystallography, the isolated CTD by NMR, and the full-length structure by SAXS. Through careful structural analysis, comparison and simulation, they identified the underlying basis for actin resistance of Dnase1L3, the conserved two-cation-assisted catalytic mechanism, and the important DNA-binding ability of Dnase1L3 CTD. This work significantly advance our understanding on the function of Dnase1L3 and the related proteins. The reviewer support the publication of this manuscript upon the fixing of some minor issues.

1. R206C mutation of Dnase1L3 is associated with various diseases. Via structural simulation, the authors concluded that R206C mutation might lower the stability of Dnase1L3. Could the authors

- express and purify the mutant and compare its stability with the native protein using CD spectra? If Dnase1L3 R206C mutant is stable, please compare its DNA-binding ability with the native protein.
2. CTD enhances the substrate-specific activity of Dnase1L3. Structural simulation resulted in two distinct DNA-CTD binding modes. It will be helpful to do some mutations and in vitro binding assays to clarify the two binding modes.
 3. Arg132 was proposed to play critical role in DNA binding and activation of Dnase1L3. Could the authors compare the DNA binding affinity of native Dnase1L3 and Arg132 mutant by in vitro assays?
 4. As stated in the method section, Dnase1L3 Δ CTD crystals diffracted to 1.9 Å. However, the structure was refined up to 2.2 Å. Why not refine the structure using all the diffraction data? High-resolution diffraction data could reveal more details for the structure, such as the cations.
 5. Please include the labels for the residue in Fig. 1B and Fig. 3C.

June 8, 2022

Reviewer #1:

Major issues:

1. The authors show an HSQC spectrum of the SH3 CTD and state that the spectrum provides information on structural variety. I see neither a discussion of nor a proof for that claim. This needs to be clarified.

Response: We have now uploaded the triple-resonance backbone assignments for the SH3-CTD and a figure showing the six lowest energy CS-Rosetta solutions for the CTD demonstrating the diversity of conformation for the CTD solutions.

2. The authors state that the CTD is an IDR, however, the NMR spectrum shows a chemical shift dispersion of over 3 ppm along the proton dimension. How do the authors explain this discrepancy?

Response: The SH3 portion of the CTD fusion protein is the well-ordered aspect of the HSQC figure (Figure 6B). The Dnase1L3 CTD are in a region of the spectra consistent with an IDP. The CTD resonance peaks are labeled accordingly. We have also now uploaded the data to the BMRB.

3. The MD simulations are said to start from the NMR structure of the CTD calculated by CS-Rosetta. There is no information the structure refinement, e.g., how many NOE constraints were used, or only chemical shift data? How reliable is the model?

Response: To address this particular concern, we added a new supplemental figure of top 6 CS-Rosetta solutions (Fig S9). This may address the issue of model reliability.

4. If the CTD is an IDP or adopts various structures, how can one refine a reliable NMR structure of it? The authors need to clarify how to obtain this structure.

Response: The point here is not to define a "high-resolution" structure or even a structural ensemble of the CTD. Instead, the CS-ROSETTA models serve as another means to show that the CTD is unstructured, which is different from what was previously proposed for this domain.

5. Also they might want to calculate SSP scores from the 13C chemical shifts in their HNCACB experiment to underline the claim that the protein adopts an intrinsically disordered structure.

Response: This is a very good point, we have included a new supplemental figure to highlight the SSP scores based on chemical shift data (Fig. S9).

6. I would also show or deposit the NMR data somewhere, as most groups do by now.

Response: We have deposited the NMR data to the BMRB with entry ID 51384. We appreciate the suggestion.

7. The main text states that the MD simulation ran 1 microsecond (also Figure S9), but the data in Figure S5 shows only 500 ns. Why?

Response: We've clarified the wording to make more explicit that the CTD and dsDNA trajectory was calculated for 1 microsecond while the R206C simulation was calculated for 500 ns.

8. Continuing with Figure S5: The differences in binding energy between the WT and the R206C mutant are on average 14 kcal/mol. This corresponds to ca. 1/3 of a hydrogen bond. How significant is that? The scatter/box plots in panel B don't look very significant to me, despite a calculated p-value of $5e-4$. Maybe the authors could comment on how they calculate this value?

Response: Sorry for any confusion. We are referring to the 3-5 kcal/mol due to salt-bridge stabilization from our reference Anderson, 1990; this corresponds to 12-20 kJ/mol. In our manuscript, we present data in kcal/mol, consistent with the units of the Rosetta Energy function used to calculate the Gibbs free energy of folding. We are aware of other estimates for salt-bridges, in part due to the effect of the local protein dielectric environment on experimental determinations of salt-bridge formation. But we deem the 3-5 kcal/mol to be in-line with standard estimates for salt-bridge contributions to free energy of folding, for example White, 1996 (doi: 10.1073/pnas.93.7.2985).

9. In Figure S9: I don't understand why the trajectory in panels A and B are traced for 12 ns, while the final structures are shown after 1 μ s? From Figure S5, it appears that the first ca. 20 ns are still subject to very strong structural fluctuations, since the system isn't yet energetically equilibrated.

Response: We have updated the text to introduce the CTD trajectory trace more clearly. The goal of this figure is to demonstrate that although the CTD to DNA binding simulation begins with the molecules spatially separated within the first 12 ns, the molecules are bound together due to rapid affinity between the CTD and dsDNA. The CTD is still experiencing strong fluctuations for the early phase of the simulations, but the rapid attraction towards the DNA target is remarkable.

10. Relating to the above I couldn't find any methods details on the MD simulations. (equilibration, solvent conditions, cut-off length etc.)

Response: It does seem the MD methods section was lacking. We have updated the methods to be more comprehensive. We greatly appreciate the critique.

Reviewer #2:

Major issues:

1. R206C mutation of Dnase1L3 is associated with various diseases. Via structural simulation, the authors concluded that R206C mutation might lower the stability of Dnase1L3. Could the authors express and purify the mutant and compare its stability with the native protein using CD spectra? If Dnase1L3 R206C mutant is stable, please compare its DNA-binding ability with the native protein.

Response: We appreciate this insight. During the months since the first review, we attempted to purify the R206C mutant of Dnase1L3 several times; however, we were unsuccessful. We attributed the difficulty in purifying mutant Dnase1L3 to its inherent lack of stability. However, we were able to express and purify Dnase1L3 R206C as an MBP-fusion protein. To experimentally verify the misfolding of the R206C mutant, even without more precise methods such as CD spectroscopy, or DNA binding affinity, we performed ANS fluorescence analysis on the MBP-Dnase1L3 R206C fusion protein and compared its interaction with ANS with MBP-Dnase1L3 WT fusion protein. The results of the MBP-Dnase1L3 fusion proteins are included in the experimental results as a new figure, Figure 2. Thank you again for the insight, we believe it adds greatly to our treatment of the R206C pathogenic mutation.

2. CTD enhances the substrate-specific activity of Dnase1L3. Structural simulation resulted in two distinct DNA-CTD binding modes. It will be helpful to do some mutations and in vitro binding assays to clarify the two binding modes.

Response: This is a very good idea we are looking forward to addressing this criticism going forward. However, we believe that purifying and performing the experimental binding assays to characterize the residue specific contributions of the CTD to DNA binding, is beyond the scope of this particular manuscript.

3. Arg132 was proposed to play a critical role in DNA binding and activation of Dnase1L3. Could the authors compare the DNA binding affinity of native Dnase1L3 and Arg132 mutant by in vitro assays?

Response: We expressed and purified Dnase1L3 R132A and performed DNA affinity assays using fluorescence polarization. We demonstrate that there is a reduction in DNA affinity with the introduction of the R132A mutation in accordance with previous results.

4. As stated in the method section, Dnase1L3 Δ CTD crystals diffracted to 1.9 Å. However, the structure was refined up to 2.2 Å. Why not refine the structure using all the diffraction data? High-resolution diffraction data could reveal more details for the structure, such as the cations.

Response: We have updated the methods to explain that we processed the experimental X-ray data to a resolution range consistent with high diffraction data completeness and de rigueur data quality standards as measured by the CC1/2 Pearson correlation coefficient. This led us to a refined structure at 2.2 Å.

5. Please include the labels for the residue in Fig. 1B and Fig. 3C.

Response: We have updated the figures with residue labels for better clarity and presentation.

REVIEWERS' COMMENTS:

Reviewer #1 (Remarks to the Author):

The authors have responded to all my questions.

Reviewer #2 (Remarks to the Author):

The authors have fully addressed my concerns and I support the publication of the revised manuscript.